# Discrete Event Simulation Model for Cost-Effectiveness Evaluation of Screening for Asymptomatic Patients with Lower Extremity Arterial Disease

**DOI:** 10.3390/ijerph191811792

**Published:** 2022-09-19

**Authors:** Vojtěch Kamenský, Vladimír Rogalewicz, Ondřej Gajdoš, Gleb Donin

**Affiliations:** Department of Biomedical Technology, Faculty of Biomedical Engineering, Czech Technical University in Prague, 272 01 Kladno, Czech Republic

**Keywords:** lower extremity arterial disease, ankle-brachial index, screening, cost-effectiveness analysis, discrete event simulation

## Abstract

Lower limb ischemic disease (LEAD) affects a significant portion of the population, with most patients being asymptomatic. Patient screening is necessary because LEAD patients have an increased risk of occurrence of other cardiovascular events and manifestations of disease, in terms of leg symptoms such as intermittent claudication, critical limb ischemia, or amputation. The aim of this work was to evaluate the cost-effectiveness of screening using ABI diagnostics in asymptomatic patients and its impact on limb symptoms associated with LEAD. A discrete event simulation model was created to capture lifetime costs and effects. Costs were calculated from the perspective of the health care payer, and the effects were calculated as QALYs. A cost-effectiveness analysis was performed to compare ABI screening examination and the situation without such screening. A probabilistic sensitivity analysis and scenario analysis were carried out to evaluate the robustness of the results. In the basic setting, the screening intervention was a more expensive intervention, at a cost of CZK 174,010, compared to CZK 70,177 for the strategy without screening. The benefits of screening were estimated at 14.73 QALYs, with 14.46 QALYs without screening. The final ICER value of CZK 389,738 per QALY is below the willingness to pay threshold. Likewise, the results of the probabilistic sensitivity analysis and of the scenario analysis were below the threshold of willingness to pay, thus confirming the robustness of the results. In conclusion, ABI screening appears to be a cost-effective strategy for asymptomatic patients aged 50 years when compared to the no-screening option.

## 1. Introduction

Peripheral artery disease (PAD) is a term used for the partial or complete occlusion of one or more artery [1]. The occlusion of an artery can be caused by a variety of factors, but the most common reason is atherosclerotic artery disease. This disease can affect various arteries, but this work focuses on lower extremity artery disease (LEAD) [1]. Clinical manifestations of the disease in lower limbs can range from an asymptomatic state to intermittent claudication (IC) or ischemic resting pain (CLI), ulceration, and gangrene [2].

Lower limb ischemic disease affects a significant portion of the elderly population, with the prevalence of the disease rising sharply with age. At ages over 50 years, it is in the range of 3–10%, with an increase to 15–20% in the population over 70 [3]. Between the years 2000 and 2010, the number of people living with PAD in developed countries increased by 13.1%, and in developing countries by 28.7% [4]. Cirgui et al. [4] stated that, despite the high prevalence of the disease and the effects of the disease on morbidity and mortality, awareness of LEAD is still limited, which may result in insufficient care or delayed treatment. In particular, patients with diabetes mellitus may be at risk of under-diagnosis, due to frequent atypical symptoms, and the presence of diabetic neuropathy may mask LEAD symptoms [5]. This has an impact not only on patients’ health but also on health systems.

According to the 2016 American College of Cardiology (ACC) and American Heart Association (AHA) Lower Extremity PAD Guideline (ACC/AHA) [6], most patients are asymptomatic, experiencing no symptoms of intermittent claudication or ulceration or gangrene in their limbs. The reported ratio between symptomatic and asymptomatic forms is 1:3–4 [3]. Those patients without classic manifestations have the same functional disability as patients with claudication [6]. Patients with LEAD (both symptomatic and asymptomatic) are at a higher risk of mortality, limb amputation, and other consequences compared to patients without LEAD. Diehm et al. [7] stated that, to reduce these risks in patients with LEAD, it is appropriate to apply early measures in the form of drug treatment. It is therefore necessary to correctly identify patients with LEAD, especially in the asymptomatic stage of the disease.

Based on the recommendations of the AHA/ACC [6], as well as the European Society of Cardiology (ESC) and the European Society for Vascular Surgery (ESVC) guidelines (ESC/ESVC) [8], an examination for suspected LEAD by a general practitioner is indicated on the basis of IC occurrence, or for patients with increased risk of LEAD. The criteria for screening are listed in Table 1. This modeling study is focused on asymptomatic patients (Fontaine classification I) and evaluation of the effectiveness of screening using the ankle-arm index (ABI).

Most modeling studies evaluating screening of LEAD with ABI have focused on the occurrence of other cardiovascular diseases, such as myocardial infarction or stroke. McDermott et al. [9,10] report that patients with LEAD, not only have increased all-cause mortality and cardiovascular events, but also show a decrease in their functional status. Likewise, McDermott et al. [11] stated that if screening is delayed until the patient is symptomatic, there is a functional decline that may be irreversible. Therefore, this work focuses on modeling LEAD in terms of disease progression to higher stages, such as IC, CLI, with differentiation of quality of life for each stage, according to Fontaine’s classification [12]; while it does not focus on the occurrence of other cardiovascular diseases, where the cost-effectiveness of screening is already supported by several studies.

Although screening examinations incur additional costs, in the long term they bring positive effects in the form of a better patient prognosis, reduced mortality, and/or a longer time spent in a better quality of life. It is appropriate to assess the impact of ABI screening on health system costs using methods of health technology assessment (HTA). The aim of this work was to evaluate the cost-effectiveness of screening using ABI diagnostics in patients with femoral artery disease in asymptomatic patients and its impact on the limb symptoms associated with LEAD.

## 2. Materials and Methods

A model was created as a discrete event simulation (DES), which, unlike Markov models, allows better simulation of the occurrence of events over time, and also allows simulating patient characteristics, which then affect the patient’s passage through the model. DES is a modeling method that, thanks to its flexibility, enables simulating complex processes. In a simulation, time moves forward at discrete intervals (from the time of one event to the time of another event) and events are mutually exclusive (events are discrete) [13]. The model was created in the R programming language [14] using a package for discrete simulations and other extension packages for probability distribution and the simulation result analysis. A list of the packages used is presented in Appendix A.

In the baseline scenario, 50-year-old asymptomatic patients with PAD and a weight of 70 kg were considered. The size of the simulated population was set on the basis of the disease prevalence for this age group, the number of inhabitants in the Czech Republic for this age group, and the ratio of men and women according to data of the Czech Statistical Office (CSO) [15]. In the model, two identical patient populations were simulated using the same pseudorandom number, and the difference in model settings depends on whether asymptomatic patients undergo PAD screening with ABI measurements or not. During the simulation, patients may experience symptoms such as intermittent claudication (disease levels IIa or IIb) or critical limb ischemia (disease levels III or IV), when undergoing various diagnostic examinations and/or therapeutic interventions, and limb amputation or death may occur. Costs and effects (Quality Adjusted Life Years; QALYs) were simulated over a lifetime horizon.

In the model, data for femoro-popliteal vessels were considered, and data specific to the superficial femoral artery (SFA) were used if available. The model did not evaluate the occurrence of the disease in other limb areas or the occurrence of the disease in the contralateral limb.

The structure of the model was designed based on the 2017 EXC/ESVC guidelines [8] for the diagnosis and treatment of peripheral arterial diseases and the guidelines from 2016 of ACC/AHA [6]. The structure of the model is shown in Figure 1, and the brief description of states in Table 2 gives the characteristics of each state.

The following diagnostic modalities were considered in the model: measurement of ankle brachial index (ABI) for disease screening, and duplex ultrasound scanning (DUS), digital subtraction angiography (DSA), computed tomography angiology (CTA), and/or magnetic resonance angiography (MRA) for other follow-up patient diagnoses. An overview of the considered sensitivity and specificity values are given in Table 3, together with their sources and the parameters of distribution used in the sensitivity analysis.

The following treatment interventions were considered in the model: exercise therapy (EXR), endovascular interventions performed using percutaneous transluminal angioplasty (PTA), PTA with stenting (PTA/S), and surgical intervention—bypass surgery. The model distinguishes whether the bypass was performed using an autologous vessel or whether an artificial vascular prosthesis was used.

Data for individual interventions were simulated separately for the patient populations with APAD, IC, and for CLI if appropriate clinical evidence was available. If evidence was not available separately for IC and CLI, the same values were considered without dividing into subpopulations (for more information on model settings, see the Appendix A).

### 2.1. Screening Settings for Patients with PAD

This study compared situations where asymptomatic patients were screened by ABI, and situations where asymptomatic patients were not screened by ABI.

The model simulated the detection and examination of patients using ABI as a part of regular examinations by general practitioners for adults, which should be performed every two years in the Czech Republic (on the basis of the Decree of the Ministry of Health of the Czech Republic No. 70/2012 Coll. on Preventive Examinations).

According to information from the Ministry of Health of the Czech Republic, 10% of the population does not attend regular check-ups, and 25% of the population attend check-ups irregularly [17]. Based on the recommendations of the AHA/ACC [6], as well as the recommendations of the ESC/ESVC [8], an ABI examination is performed by physicians when PAD is suspected and there is an increased risk of PAD. The first two categories of groups of patients at increased risk of PAD are considered in the model (see Table 1).

Patients who regularly undergo preventive check-ups are examined using a non-invasive ABI examination if at least one of the first two conditions listed in Table 1 is met. If the patient does not meet the conditions for the examination, the time until the next examination is simulated (based on the recommended frequency of examinations every two years). Patients irregularly undergoing preventive check-ups are considered to have a 50% chance at the time of a preventive check-up (no more information was found on the frequency of irregular check-ups) of whether they will attend for the check-up or not (the model logic is presented in the diagram on Appendix A).

If the examination diagnoses PAD, the patient is indicated for pharmacological treatment. Patients with PAD can be treated with a wide variety of possible medications. As no study evaluating the effect of the therapy mix in patients with PAD without intermittent claudication has been found, the same effect is expected as in patients with IC. The effect of a possible drug combination may vary, and therefore the expected effect in the model is simulated as an adjustment for death and limb symptom occurrence, applying a hazard ratio (HR) of 0.75 (HR was estimated based on the studies by Golledge and Drovandi [18] and Bevan and Khendi [19]).

If further stages of the disease occur within the simulation, immediate diagnostic and therapeutic care is assumed in the model (regardless of whether the patient attended preventive check-ups or not), and pharmacological treatment is indicated.

In the scenario without ABI screening, patients are diagnosed and treated only after the transition from an asymptomatic state to the state of IC, CLI, or limb amputation.

### 2.2. Follow-Up Diagnostic and Therapeutic Care

The subsequent simulation is the same for all evaluated scenarios. If a patient develops intermittent claudication, he/she is first indicated for exercise treatment, regardless of whether he/she attended regular check-ups or not. If the condition worsens in the CLI during the simulation, the patient is sent for further diagnosis and the corresponding intervention therapy is indicated (the model logic is presented in the diagram of Appendix A). The patient is indicated for further diagnostic examination and subsequent therapy if he/she is of the grade IIb (disease status IC) and he/she does not improve after two completed therapies with exercise (thus, the patient has a lifestyle-limiting claudication). The type of diagnostic imaging examination and the type of interventional therapy are assigned to the patient randomly.

If the patient is indicated for interventional treatment, it is assumed that all patients are suitable for the first intervention. For intervention treatment, the possibility of death during therapy (30-day mortality), and also the occurrence of complications (30-day morbidity), are simulated, and the occurrence of complications will affect the value of the intervention costs. For endovascular interventions, the technical success is simulated, where in the case of a failed intervention, the patient may undergo another intervention one year after the failed intervention. In the model, a patient can undergo two successful interventions; if he/she has two unsuccessful interventions, the simulation assumes that he/she can no longer undergo any further interventional treatment.

If the patient successfully undergoes interventional treatment, the time of patency of the vessel is simulated. As long as patency is maintained, the patient is in an asymptomatic state (shown by the dotted line in Figure 1); after losing patency, the patient returns to the state before the intervention. Depending on whether the patient is undergoing their first or second intervention therapy, different probability distributions for simulation of patency times are used (primary or secondary patency).

In case of a loss of patency or a technical failure of endovascular interventions, the patient is indicated for reoperation. For each reoperation, the probability that the patient is not suitable for reoperation is simulated (values different for IC and CLI). This approach is consistent with that of Simpson et al. [20]. The model logic is presented in the diagram in Appendix A.

If the patient’s limb is amputated during the simulation, he can no longer undergo interventional therapy and remains in an amputation state until his/her death. The model distinguishes whether there is an amputation below (BKA) or above (AKA) the knee joint. The mortality after limb amputation is simulated, as well as the rate of complications (based on DRG reported data [21]).

The health condition affects patient’s time to death, while the interventions that the patient undergoes do not affect the time to death. This assumption was determined based on the study by Simpson et al. [20]. In order to maintain consistency in patient survival when events affecting patient survival occur, the procedure proposed by Corro Ramos et al. [22] was used (e.g., that the survival of patients does not increase when IC occurs). All model structure setting values are listed in the Appendix A.

### 2.3. Utility

Utilities associated with each state were taken from published literature. Studies evaluating quality of life using the generic EuroQol EQ-5D questionnaire were used as a source of utilities. An overview of the considered utility values for each state is given in Table 4.

### 2.4. Costs

The costs are considered in the model from the perspective of the health care payer for 2021; in the Czech Republic these are health insurance companies. The values of the considered costs are listed in Table 5. The costs of preventive examinations by general practitioners, outpatient specialists, diagnostic examinations using diagnostic modalities, and outpatient exercise therapy were determined on the basis of the list of medical services and reimbursement principles specified in the Decree of the Ministry of Health of the Czech Republic No. 134/1998 Coll., which states the list of medical services with point values [26]. Reimbursements are considered using the point values for 2021. For CTA, MRA, and DSA examinations, the use of a contrast agent is required, therefore the costs of the contrast agent according to the considered volume for a 70 kg patient was added to each operation [27].

In the case of conservative exercise therapy, the patient is considered to undergo the therapy in an outpatient setting. Based on the recommendations of AHA/ACC [6], a 12-week therapy with 45-min exercise blocks three times a week is considered. Thus, there are 36 exercise units in total, to which the costs of the initial examination and the costs of the final evaluation of the therapy by the rehabilitation physician are added.

The costs of interventional treatment of PTA, PTA/S or bypass, as well as the costs of limb amputation, were determined on the basis of the reported average costs using DRG [21]. Possible reimbursement codes and average reimbursements of insurance companies for these interventions were found for each intervention. Based on the simulation of possible complications during interventional therapy (30-day morbidity), different DRG codes were considered, which capture the higher cost of more complicated cases. For limb amputation, different DRG codes and costs are considered in the model, depending on the type of hospital (e.g., Center for highly specialized care; CHSC) where the amputation is performed, and according to the type of amputation (AKA is defined as severe).

After amputation of the limb, it is necessary to take care of the amputated limb and patients need additional follow-up treatment in the form of rehabilitation and prosthetic care, which is divided according to the type of amputation. These costs are considered in the model as annual costs (rehabilitation costs as annual costs were only considered one year after amputation).

The model assumes that patients with PAD without manifestations of IC or CLI are treated by outpatient general practitioners (PAD screening). In the case of IC or CLI symptoms, patients are monitored on an outpatient basis in an angiology outpatient clinic (indications for examination or treatment), and patients undergoing bypass surgery are examined by a vascular surgeon before and after surgery.

For stage IV disease, the annual cost of treating ulceration and gangrene is considered. Patients diagnosed with PAD are prescribed pharmacological treatment, the costs of which are considered in the model as annual costs.

### 2.5. Model Validation

As part of the creation of the model, the face validity, internal validity, and cross validity were assessed following ISPOR-SMDM recommendations [33]. Face validity was verified based on a consultation with a clinician having expert knowledge in the field of cardiology. The structure of the model was presented to clinicians along with the assumptions used. Furthermore, the structure of the model was compared with the structure and assumptions of other models focused on LEAD modeling. To determine the appropriateness of the aim of the study, the use of modeling techniques and type of cost analysis were validated with experts in the field of health technology assessment. Scripts for simulation of cost effectiveness, and for processing and presentation of results, were verified (assessment of internal validity) by a co-author of the work who was not involved in the coding and is experienced in working with the R program. Cross validity was verified by comparing the structure of the model with other models focused on LEAD.

### 2.6. Cost-Effectiveness Anlysis

The cost-effectiveness analysis is presented in the form of average patient costs and average QALYs gained. The decision on the cost-effectiveness of screening is made on the basis of an incremental cost-effectiveness ratio (ICER). The value of ICER is compared with the threshold of willingness to pay, which in the Czech Republic is set at CZK 1.2 million per QALY, according to the recommendations of the State Institute for Drug Control (SUKL) [34]. Alternatively, the value of three times the gross domestic product per capita (GDP per capita) can be used, which for the year 2021 was CZK 1,748,610 per QALY.

### 2.7. Sensitivity Analysis

A sensitivity analysis was performed based on the recommendations of ISPOR-SMDM [35]. A probabilistic sensitivity analysis was performed as the main analysis. In the sensitivity analysis, 1000 simulations of both scenarios were first performed, using different pseudo-random numbers. In addition, 1000 simulations were performed using the same pseudo-random number as in the base case scenario, but the input parameters were defined using the probability distributions listed in the model input tables (Table 3, Table 4 and Table 5) and the Appendix A.

### 2.8. Scenario Analysis

As part of the scenario analysis, the values of average costs, average QALYs, and ICER were also calculated for the model setup: (i) all patients go for regular check-ups during screening; (ii) sensitivity value for ABI 0.90 (based on values used by Vaidya et al. [36]); (iii) a discount rate of 5%; and (iv) time between next ABI examination 3, 4, and 5 years.

## 3. Results

In the basic setting of the model, 8843 asymptomatic patients with PAD were simulated. The lifetime horizon simulation calculated the total average patient treatment costs and the total average QALYs (results with discount rates of 3% and 0% are presented in Table 6).

The screening scenario was the more costly intervention in both cases (CZK 174,010 vs. CZK 70,177 and CZK 268,708 vs. CZK 112,096, respectively) but generated multiple effects in the form of QALYs (14.73 vs. 14.46 and 21.08 vs. 20.60, respectively). Considering the discount rate of 3% (both in costs and effects), the resulting ICER values (389,738 and 320,396 respectively) were below the WTP threshold, both when considering the SUKL value MCZK 1.2 per QALY. Screening was thus the more cost-effective intervention. Cumulative costs (Part A) and QALYs (Part B) are shown in Figure 2. Due to the longer survival of patients who undergo pharmacological treatment after screening, more effects are generated in the form of QALY, but also higher costs are gradually accumulated, including the cost of pharmacological treatment, with an increasing number of patients receiving follow-up diagnostics and treatment.

When analyzing deaths from any cause, the resulting simulated median survival in the screening scenario was 24.6 years (95% CI 24.3, 24.8 years), and in the no-screening scenario this was 23.7 years (95% CI 23.3, 23.9 years).

### 3.1. Results of Sensitivity Analysis

The uncertainty associated with the cost-effectiveness estimate is shown in Figure 3 and in the Appendix A. Figure 3 shows the results of probabilistic analysis in the simulation of input parameters using defined probability distributions. Part A in Figure 3 shows the cost-effectiveness scatter plane with the results of individual ICER values from individual iterations (red dots in Figure 3). All points are in the upper right quadrant; screening is always a more costly, but also more effective, intervention. All ICER values are below the WTP limit and therefore screening is a cost-effective intervention in all cases. The average mean value of the cost difference and the effect difference at a discount rate of 3% was CZK 104,487 and 0.32, respectively, and the resulting ICER value was CZK 325,440 per QALY. Part B shows the cost-effectiveness acceptability curve, where it can be seen from the WTP greater than CZK 450,000 that the ICER of all iterations were cost-effective.

### 3.2. Results of Scenario Analysis

When considering the model setting, where all patients regularly attend preventive check-ups, the average costs were CZK 190,973 and QALYs 14.79, giving a final ICER value of CZK 369,647 per QALY. Even in this case, screening is a cost-effective intervention. If the ABI sensitivity value was set to 0.9 and the other parameters were the same as in the basic setting, then for the screening scenario the average cost was CZK 178,484 (no screening CZK 69,448) the average gain was 14.74 QALYs (no screening 14.47 QALYs), and the resulting ICER value was CZK, 386,496 per QALY. Screening remains a cost-effective intervention. In the last simulated scenario with a 5% discount rate, the average cost per scenario with screening was CZK 136,546, and without screening it was CZK 53,695. The average QALYs were 12.11 for screening and 11.93 for no-screening. The resulting ICER value of CZK 452,980 per QALY remained below the WTP threshold in this case as well.

## 4. Discussion

The diagnosis of LEAD in asymptomatic patients is most often determined on the basis of the measurement of ABI value. The study by Hirsch et al. [37] states that physicians who use only reported IC for detection of LEAD are unlikely to diagnose 85–90% of LEAD patients. Sutkowska et al. [38] pointed out that there is a low awareness of patients, regarding the possible manifestations of LEAD, and that even medical staff appear to give the minimum attention to some examinations such as, for example, foot examination. The ABI value indicates the ratio between systolic blood pressure measured at the ankle and systolic blood pressure measured at the brachial artery [39]. Decreased ABI signifies hemodynamically significant occlusive disease between the heart and ankle [3]. The ankle-arm index has a good validity and thus it is recommended as the first-choice test for screening and diagnosing LEAD, as well as in patients with a history or physical examination indicating LEAD [6]. Johnston et al. [40] stated that abnormal ABI values are commonly associated with a greater risk of MI, stroke, and other cardiovascular events, but are not consistently associated with patient-centered assessment of functional or health status.

Similarly, the published models [36,41,42] focused more on modeling the occurrence of other cardiovascular events after screening than on the modeling of limb symptoms of LEAD. Therefore, to evaluate the cost-effectiveness of the screening examination using the ABI method, a DES model simulating peripheral arterial disease was created. In contrast to other models [36,41,42], this model tries to capture the development of individual symptomatic phases of LEAD, such as IC and CLI.

More severe forms of LEAD are associated with a higher degree of functional impairment in patients [11]. Detection of patients with LEAD, even without limb symptoms, is important. McDermott et al. [11] found that patients with low ABI values had a significantly greater decrease in walking abilities after two years of follow-up. Patients with ABI values of 0.50 were up to 13-times more likely to be unable to walk for more than 6 min after two years. A decrease in walking ability need not be observed only in patients with a significant decrease in ABI, asymptomatic patients also have a significantly greater decrease in walking ability compared to patients without LEAD, while patients with borderline values and normal values (range 1.00–1.09) already have a significantly higher loss of mobility than patients with normal ABI values in the range of 1.10–1.30 [11,43].

The developed model also differs in its approach to disease modeling. In contrast to other models evaluating the cost-effectiveness of ABI screening, the model was created as DES, instead of using Markov models together with decision trees. According to the literature (e.g., [44]), DES models can be more difficult to create, but due to the simulation of individual patient characteristics, while the occurrence of events is simulated as the time of occurrence of the event, they may more accurately reflect the natural course of the disease. In Markov models, there is a transition of patients between individual states with a certain probability in cycles.

The structure of this model is similar to the model of Simpson et al. [20], which also used the DES modeling technique, but did not focus on an evaluation of disease screening and simulating asymptomatic patients. The simulation of asymptomatic patients is similar to other screening evaluation models. The model differs from models published by Vaidya et al. [36], Itoga et al. [41], and Lindholt and Søgaard [42] in some settings, such as the considered age of patients or the number of possible repeated screening examinations using ABI. Comparisons with other models are further described further in the discussion.

No other DES model evaluating the cost-effectiveness of ABI screening were found in the literature. However, the DES modeling technique was used by Simpson et al. [20] to evaluate the cost-effectiveness of interventional therapy (PTA, PTA/S and bypass).

At the baseline settings, the model compared the PAD screening in asymptomatic patients with the situation where no screening is performed. Similarly to the study by Vaidya et al. [36], and based on the recommendations of professional societies, we simulated screening using the ABI method performed by general practitioners.

The analysis was performed from the perspective of the health care payer. In all model settings, screening was a more expensive intervention, with higher effect values in the form of QALYs. The results of the basic model setup, as well as the results obtained from the sensitivity analysis and scenario analysis, showed that ICER was always below the WTP threshold. Thus, the results of cost-effectiveness modeling suggest that despite higher overall costs, screening is a cost-effective strategy, due to a better patient survival and the associated longer life expectancy, which has an impact on achieving more QALYs. In the basic setup of the model, the average cost of the screening reached CZK 174,010, and the strategy generated an average of 14.73 QALYs in the lifetime horizon. Compared to the strategy with no screening, the costs were higher by CZK 103,834, and the effects were higher by 0.26 QALYs. The value of ICER was CZK 389,738 per QALY for the screening option, which is below the WTP threshold (MCZK 1.2 per QALY in Czechia). Although a different modeling technique was used and the model simulated the development of the disease, the conclusions that screening is a cost-effective strategy are in accordance with the conclusions of other authors. The probabilistic sensitivity analysis confirmed the robustness of the results.

Aboyans et al. [39] mentions that it would be appropriate to evaluate, from a cost-effectiveness perspective, how often ABI measurements should be repeated. In asymptomatic patients, due to the good sensitivity of the ABI measurement, a longer interval between repeated measurements does not have a significant effect on the final ICER value. The time between repeated measurements may have a more significant effect on cost-effectiveness when simulating patients with and without LEAD.

Three studies were found that modelled the cost-effectiveness of LEAD screening. Vaidya et al. [36] developed a probabilistic model to evaluate the lifetime cost-effectiveness of single-screening and follow-up treatment (aspirin and clopidegrel are considered in the model). The model is a combination of a decision tree for evaluating the success of screening and a Markov model for simulating the subsequent risk of cardiovascular events. Vaidya et al. [36], thus, did not simulate the disease to more burdensome stages, as well as omitting a possible subsequent use of a wide range of different diagnostic and therapeutic methods for the treatment of limb symptoms. Vaidya et al. [36] calculated costs and effects over a lifetime horizon, but considered a societal perspective for costs. The considered population also differed slightly, where the initial age of men and women was 55 years. In addition, in our study, we considered the possibility of repeated screening using the ABI method, performed during preventive examinations.

The resulting lifetime QALY values are comparable to those obtained in this study. The life-years gained are lower in the Vaidya et al. [36] study, but this is probably due to the higher starting age of the cohort of patients. The conclusion of Vaidya et al. [36] was that ABI screening is the dominant intervention over a situation without screening. Probabilistic sensitivity analysis showed that more than 88% of iterations were in the lower right quadrant of cost-effectiveness, therefore remaining the dominant intervention. In a deterministic sensitivity analysis, they identified disease prevalence and the effect of low doses of aspirin (in the form of RRs) as the most influential parameters.

Itoga et al. [41] also used a combination of Markov models and decision trees to evaluate the cost-effectiveness of screening of asymptomatic patients using ABI. Similarly to Vaidya et al. [36], they considered the progression of the disease to symptomatic PAD and the occurrence of cardiovascular events in their model. In contrast to Vaidya et al. [36] and the set-up of this study, they considered a population of healthy patients and the initial age of patients was 65 years. Their model worked with a horizon of 35 years. Itoga et al. [41] did not distinguish between different symptomatic LEAD states (such as IC and CLI), but considered the possibility of endovascular or surgical treatment in symptomatic patients. They showed that screening is a more expensive strategy, but similar to the results of our study and that of Vaidya et al. [36] there was a greater QALY gain. The resulting ICER value of USD 88,758 per QALY is within the cost-effectiveness limit (USD 50,000–100,000). It should be noted that Itoga et al. [41] stated that there was insufficient empirical evidence regarding the effect of ABI screening on medication and reducing the risk of further events, and that caution is needed in interpreting the resulting findings.

Lindholt and Søgaard [42] also used a Markov model to evaluate screening in men. If available, the source data were based on the VIVA trial [45]. The authors used a model suggested by Vaidya et al. [36], but they also considered a state without LEAD. In the model, they simulated the lifetime horizon with a patient starting age of 65 years. Consistently with the results of this study, screening increased patient survival and there was a higher gain in QALYs. Consistently with the results of this study and the results of Itoga et al. [41], and in contrast to the results of Vaidya et al. [36], the screening scenario was more costly. According to the authors, the resulting ICER value of the baseline scenario of EUR 20,673 per QALY can be considered a cost-effective strategy, which is in accordance with conclusions of the other authors [36,41].

In the context of the COVID-19 pandemic, the model was used to estimate life-years and QALYs lost due to a decrease in preventive check-up visits over the two years of the pandemic. According to the Ministry of Health of the Czech Republic, the reporting of preventive examinations by general practitioners decreased by almost 14.6% [46]. The proportion of patients who attend regular check-ups, attend check-ups irregularly, or those who do not attend check-ups were adjusted to reflect this decrease (55.64%, 23.20%, and 21.16%, resp.). The model was set in such a way that during the first two years of the simulation, the proportion of people attending preventive examinations changed, and in the remaining simulation time, the model was set as in the basic settings. In the simulated population of 8843 patients (fifty-year-old), there was a total loss of 480 years of life and 74 QALYs, just from the two years of decreased preventive examinations.

### Limitations and Future Research Directions

Anatomical parameters (e.g., lesion length, stenosis size, etc.) are not simulated in the model. Although these characteristics can affect the diagnosis and treatment of the disease, due to a lack of quality evidence, these parameters were not used. These limitations are in accordance with the approach of Simpson et al. [20]. Moreover, no model simulating these parameters was found.

A limitation (in comparison to the Vaidya et al. [36], Itoga et al. [41], and Lindholt and Søgaard [42] studies) is the omission of modeling the occurrence of other possible cardiovascular events, but the model captures a higher mortality of patients with PAD compared to the general population, as well as an increased risk of mortality in IC and CLI forms.

Simulating the occurrence of other cardiovascular events and their impact on costs and effects may allow for extending the model, as well as modifying the model input data for a subpopulation of patients with diabetes mellitus (DM). In our model, DM is simulated as a factor influencing the indication for screening. Additional input data would be needed to evaluate screening in patients with DM, as patients with DM have a higher risk of death, amputations, and even different outcomes from interventional therapy. However, because the model was created using DES, this extension of the model was covered within the simulations of individual patients with unique characteristics and the time to events distribution for patients with DM. Likewise, the model could be extended by simulation of different types of intervention therapy, next to the considered PTA, PTA/S, and bypass. In addition to the chosen and commonly used treatments evaluated in this work, other methods could be used in various stages of the disease, to maintain the functional status of the limb, such as pulsatile pneumatic compression therapy, the systemic hyperbaric oxygen therapy, lower limb offloading, or other physiotherapy interventions [47].

This model may serve as the first tool to build a comprehensive model of diagnostic-therapeutic care for LEAD. Then, a single model could be used to solve a wide range of decision-making problems, as suggested by Tappenden et al. [48,49] and Lord et al. [50].

## 5. Conclusions

Discrete event simulation seems to be a suitable modeling technique for the evaluation of the long-term cost and health consequences of screening programs in asymptomatic patients with LEAD. This approach allows capturing LEAD, from the asymptomatic condition to the occurrence of limb symptoms IC, CLI, amputation, and patient death. Thanks to the DES modeling technique, this is a variable model that allows different model settings, in order to support decisions about cost-effectiveness. The model created could also be used for other decision problems in LEAD. When simulating lifetime costs and benefits, screening using the ABI diagnostic method appears to be a cost-effective strategy for asymptomatic patients aged 50 years, compared to no screening. It is therefore appropriate to support general practitioners in conducting ABI screening and to raise awareness of its clinical as well as cost benefits. It is also important to increase patients’ interest in preventive examinations, so that the disease is detected in the asymptotic stage.

## Figures and Tables

**Figure 1 ijerph-19-11792-f001:**
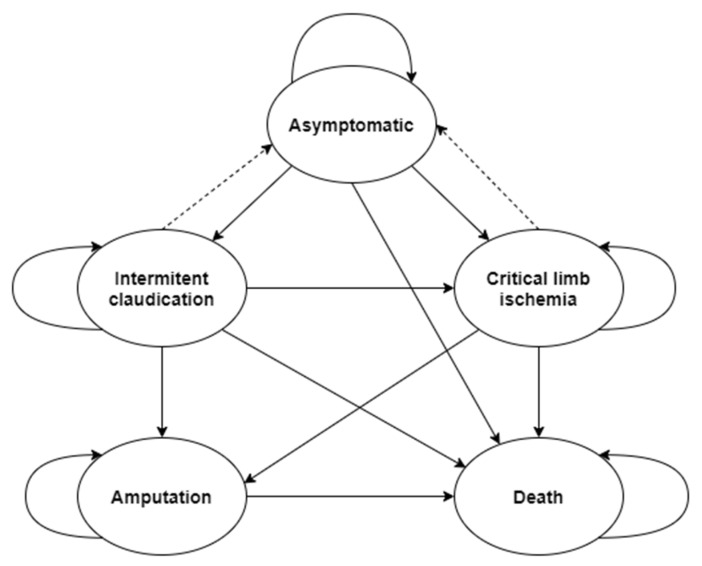
Considered states in the model and possible transitions between states.

**Figure 2 ijerph-19-11792-f002:**
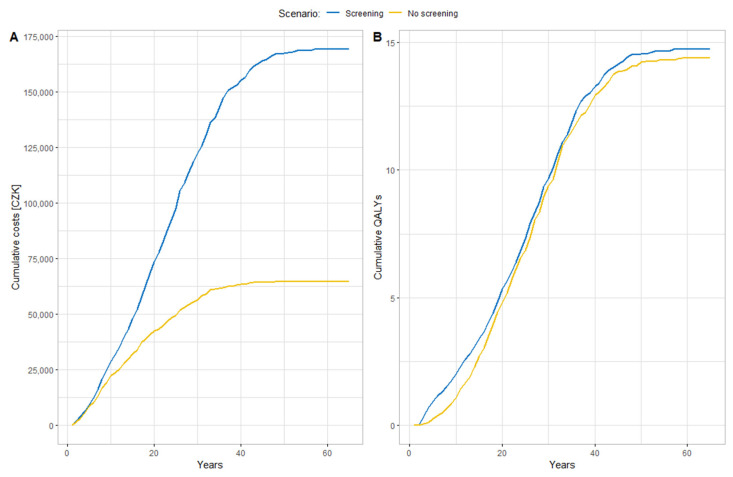
(**A**): Mean cumulative costs (CZK) per patient; (**B**): Mean cumulative QALYs per patient.

**Figure 3 ijerph-19-11792-f003:**
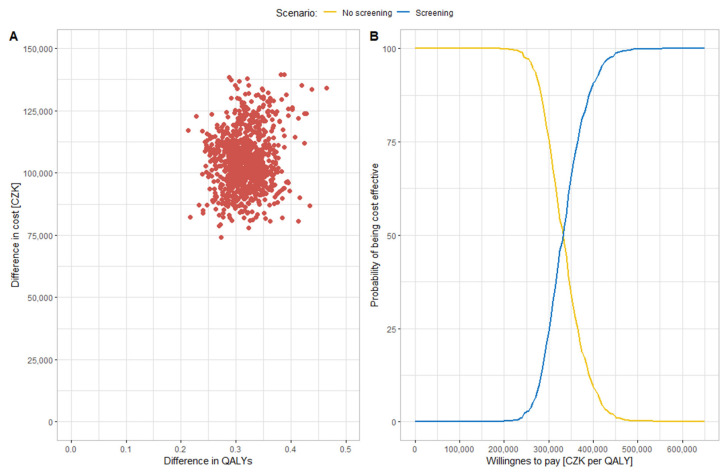
(**A**): Cost-effectiveness scatter plane with input variability; (**B**): cost-effectiveness acceptability curve for probabilistic analysis with input variability.

**Table 1 ijerph-19-11792-t001:** Patients at increased risk of PAD recommended for screening [6].

Categories of High-Risk Patients	Patients Characteristics
Category 1	Age over 65 years.
Category 2	Age between 50–64 years with risk factors for atherosclerosis (e.g., diabetes mellitus, history of smoking, hyperlipidemia, hypertension) or family history of PAD.
Category 3	Age under 50 with diabetes mellitus and one other risk factor for atherosclerosis.
Category 4	Individuals with a known incidence of atherosclerosis in another vascular area (e.g., coronary, carotid, etc.)

**Table 2 ijerph-19-11792-t002:** Description of health states considered.

State	State Characteristics
Asymptomatic	In this state, there are patients with PAD without limb symptoms (class I according to Fontaine’s classification).
Intermittent claudication	In this state, there are patients with intermittent claudication (class IIa and IIb according to Fontaine’s classification).
Critical limb ischemia	In this state, patients are of the class III and IV according to the Fontaine classification, they experience ischemic resting pain (class III), or already have ulcerations or gangrene of the limb (class IV).
Amputation	In this state there are patient after limb amputation due to PAD. Only so-called large amputations (amputations above the level of the ankles) are considered in the model.
Death	Patients who died from PAD, IC, CLI, amputation or interventional therapy are in this state.

**Table 3 ijerph-19-11792-t003:** Parameters of diagnostic modalities.

Diagnostics	Sensitivity	Sensitivity Analysis	Source
ABI	0.75	Beta ^1^ (24.25, 8.08)	[8]
DSA	0.97	Beta ^1^ (2.03, 0.06)	[16]
CTA	0.96	Beta ^1^ (3.04, 0.13)	[8]
MRA	0.96	Beta ^1^ (3.04, 0.13)	[8]
DUS	0.87	Beta ^1^ (12.13, 1.81)	[8]

^1^ parameters shape 1 and shape 2.

**Table 4 ijerph-19-11792-t004:** Utility values used in the model.

State	Utility	Sensitivity Analysis	Source
Asymptomatic PAD (age 50–64)	0.92	Beta ^1^ (7.08; 0.62)	[23]
Asymptomatic PAD (age 65–74)	0.89	Beta ^1^ (10.11; 1.25)	[23]
Asymptomatic PAD (age 75+)	0.84	Beta ^1^ (15.16; 2.89)	[23]
IC (Fontaine IIa)	0.63	Beta ^1^ (36.37; 21.36)	[24]
IC (Fontaine IIb)	0.52	Beta ^1^ (47.48; 43.83)	[24]
CLI (Fontaine III)	0.44	Beta ^1^ (55.56; 70.71)	[24]
CLI (Fontaine IV)	0.40	Beta ^1^ (59.60; 89.40)	[24]
Amputation (BKA)	0.61	Beta ^1^ (38.39; 24.54)	[25]
Amputation (AKA)	0.20	Beta ^1^ (79.80; 319.20)	[25]

^1^ parameters shape 1 and shape 2.

**Table 5 ijerph-19-11792-t005:** Cost values used in the model.

Cost	Value (CZK ^4^)	Sensitivity Analysis	Source
ABI	157	Variation of the point value; uniform distribution within ±20% interval ^3^.	[26]
DUS	1154		[26]
DSA	10,248		[26]
CTA	1585		[26]
MRA	6381		[26]
PTA	76,769	Log-normal ^2^ (11.22, 0.21)	[21]
PTA/S	110,427	Log-normal ^2^ (11.58, 0.24)	[21]
PTA, PTA/S with complication	175,580	Log-normal ^2^ (11.99, 0.20)	[21]
Bypass	138,681	Log-normal ^2^ (11.85, 0.06)	[21]
Bypass with complication	199,387	Log-normal ^2^ (12.21, 0.07)	[21]
Bypass with severe complication	322,796	Log-normal ^2^ (12.69, 0.10)	[21]
Amputation	108,936	Log-normal ^2^ (11.61, 0.04)	[21]
Amputation with complication	173,636	Log-normal ^2^ (12.07, 0.04)	[21]
Amputation in CHSC	156,315	Log-normal ^2^ (11.96, 0.05)	[21]
Amputation with complication in CHSC	268,217	Log-normal ^2^ (12.50, 0.05)	[21]
Post amputation care	66,963	Log-normal ^2^ (11.11, 0.10)	[28]
Prosthetic care AKA amputation	62,455	Log-normal ^2^ (11.04, 0.10)	[28]
Prosthetic care BKA amputation	59,465	Log-normal ^2^ (10.99, 0.10)	[29]
Prosthetic care-service	15,640	Log-normal ^2^ (9.65, 0.10)	[28]
Treatment ulceration and gangrene	46,608	Log-normal ^2^ (10.75, 0.10)	[30]
Pharmacological care	7349	Log-normal ^2^ (8.90, 0.10)	[31,32]
Examination by a general practitioner	963	Variation of the point value; uniform distribution within ±20% interval ^3^.	[26]
Repeated examination by a general practitioner	645		[26]
Comprehensive angiologist examination	872		[26]
Targeted angiologist examination	440		[26]
Control angiologist examination	221		[26]
Comprehensive vascular surgeon t examination	462		[26]
Targeted vascular surgeon examination	311		[26]
Control vascular surgeon examination	155		[26]
Examination of the claudication interval	139		[26]
Exercise therapy	12,652 ^1^		[26]
Complex examination by a rehabilitation doctor	872		[26]
Targeted examination by a rehabilitation doctor	440		[26]

^1^ Cost for complete 36 exercise unit; ^2^ parameters logmean and logsd; ^3^ parameters minimum and maximum; ^4^ average EUR/CZK exchange rate in 2021 was 25.645 and for USD/CZK was 21.682.

**Table 6 ijerph-19-11792-t006:** Results of the cost-effectiveness simulation.

Scenario	Total Cost (CZK ^1^)	Difference in Cost (CZK)	Total QALYs	Difference in QALYs	ICER (CZK per QALY)
** *Results with discount rate = 3%* **					
No screening	70,177	-	14.47	-	-
Screening	174,010	103,834	14.73	0.26	389,738
** *Results with discount rate = 0%* **					
No screening	112,096	-	20.60	-	-
Screening	268,708	156,612	21.08	0.48	320,396

^1^ Average EUR/CZK exchange rate in 2021 was 25.645, and for USD/CZK it was 21.682.

## Data Availability

All input parameters are listed in the text and in the Appendix A. Model scripts created in RStudio are available upon request from the corresponding author.

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
