# Peer review of "Discrete Event Simulation Model for Cost-Effectiveness Evaluation of Screening for Asymptomatic Patients with Lower Extremity Arterial Disease"

_ijerph, 2022, doi:10.3390/ijerph191811792_

Round 1

Reviewer 1 Report

The authors analysed the cost and effectiveness of ABI diagnostic screening in asymptomatic patients and its impact on limb symptoms associated with LEAD. A variable simulation model of discrete events in asymptomatic patients was created to capture effects and costs. 

The study is well conducted and the methods used are appropriate. The data is presented clearly.

The results demonstrate the importance of screening and the clinical as well as cost benefits of screening The authors' analysis of screening is an important and cost-effective strategy.  Screening of patients with LEAD is necessary due to disease progression and increased risk of other cardiovascular events.

This is a good analysed and written paper, with an interesting observation with potential clinical impact. I believe that this manuscript will be of genuine interest to all members of the broad community, including researchers, practitioners, and policymakers.  I would recommend publication, after the issues below are addressed. 

The introduction and discussion needs to highlight , that the problem is mainly in group of the patients with diabetes, because the neuropathy can mask the symptoms of ischaemia. Both neuropathy and ischemia are responsible for ulceration and finally amputation mainly if abnormal distribution of the plantar pressure exists.  (Sutkowska E, Sutkowski K, Sokołowski M, Franek E, Dragan S Sr. Distribution of the Highest Plantar Pressure Regions in Patients with Diabetes and Its Association with Peripheral Neuropathy, Gender, Age, and BMI: One Centre Study. J Diabetes Res. 2019 Jul 9;2019:7395769. doi: 10.1155/2019).
Unfortunatelly neuropathy as well as ischaemia are rarelly checked in everyday practice as was stressed in the studies (eg: Sutkowska E., Sokołowski M., Dragan S., and Zdrojowy K., The practitioners’ compliance in diabetic foot risk searching, „Diabetes Stoffwechsel und Herz”, 2015, t.24, no 6, Diabetes Stoffwechsel und Herz, p. 429,toż:,Vol.; Sutkowska EE, Sokołowski M, Zdrojowy K, Dragan S. Active screening for diabetic foot — assessment of health care professionals’ compliance to it. Clin Diabet 2016; 5, 3: 83–87. DOI: 10.5603/DK.2016.0014 or another paper in this topic)However for ischaemia the revascularisation is the most important line of treatment also the alternative methods should be considered and applied et each stage of the disease e.g. intermittent pneumatic compression (Sutkowska E, Wozniewski M, Gamian A, Gosk-Bierska I, Alexewicz P, Sutkowski K, Wysokinski WE. Intermittent pneumatic compression in stable claudicants: effect on hemostasis and endothelial function. 
Int Angiol. 2009 Oct;28(5):373-9. PMID: 19935591. And Hap K, Biernat K, Konieczny G. Patients with Diabetes Complicated by Peripheral Artery Disease: the Current State of Knowledge on Physiotherapy Interventions. J Diabetes Res. 2021 May 10;2021:5122494. doi: 10.1155/2021/5122494. PMID: 34056006; PMCID: PMC8131145.

Otherwise, I found the manuscript very well organized and clearly written.

Reviewer 2 Report

One of the main concerns I have with this paper is that there is no discussion on the model validity. In order to support the authors' findings, it needs to be shown that the DES model the authors develop correctly reflect the intended situations. But, this paper fails to provide the evidences for model validity. Without it, all the discussions the authors made would not be enough for the readers to accept them. 

Reviewer 3 Report

The article has a very important theoretical foundation....

Based on the review of the article, the following suggestions are made to improve some aspects:

0. Abstract: it is suggested to include the main conclusions reached in the study.

1. it is recommended to include the definition of discrete simulation.

2. Explain the difference between having used R versus a specialized software such as ProModel, among others.

3. What was the model validation process? What kind of tests were made to the current model to perform sensitivity analysis?

Line 271: The policies that were used for the sensitivity analysis are supported by the State Institute for Drug Control (SUKL)...I recommend mentioning the most relevant ones to put the reader in context.

Line 307: Figure 2 is cited but was not included in the paper, nor in the Supplementary document.

Line 321: Figure 3 is cited, but in the Supplementary document it appears as Figure S4 in the paper, nor the Supplementary document (Figure S4: Cost-effectiveness scatter plane using different pseudo-random number; B: Cost-effectiveness acceptability curve for probabilistic analysis using different pseudo-random number).
